# Co-culture models of endothelial cells, macrophages, and vascular smooth muscle cells for the study of the natural history of atherosclerosis

**Martin Liu[1]☯, Saurabhi Samant[2]☯, Charu Hasini Vasa[1,2], Ryan M. Pedrigi[3], Usama M. Oguz[1,2], Sangjin Ryu[4], Timothy Wei[4], Daniel R. Anderson[2], Devendra K. Agrawal[5], Yiannis S. Chatzizisis[1,2]***

1 Division of Cardiovascular Medicine, Miller School of Medicine, University of Miami, Miami, Florida, United States of America, 2 Cardiovascular Division, University of Nebraska Medical Center, Omaha, Nebraska, United States of America, 3 Department of Biological System Engineering, University of Nebraska-Lincoln, Lincoln, Nebraska, United States of America, 4 Department of Mechanical and Materials Engineering, College of Engineering, University of Nebraska-Lincoln, Lincoln, Nebraska, United States of America, 5 Department of Translational Research, Western University of Health Science, Pomona, California, United States of America

☯ These authors contributed equally to this work.
* ychatzizisis@icloud.com

**Data Availability Statement:** All relevant data are within the paper and its Supporting Information files.

## Abstract

### Background

This work aims to present a fast, affordable, and reproducible three-cell co-culture system that could represent the different cellular mechanisms of atherosclerosis, extending from atherogenesis to pathological intimal thickening.

### Methods and results

We built four culture models: (i) Culture model #1 (representing normal arterial intima), where human coronary artery endothelial cells were added on top of Matrigel-coated collagen type I matrix, (ii) Culture model #2 (representing atherogenesis), which demonstrated the subendothelial accumulation and oxidative modification of low-density lipoproteins (LDL), (iii) Culture model #3 (representing intimal xanthomas), which demonstrated the monocyte adhesion to the endothelial cell monolayer, transmigration into the subendothelial space, and transformation to lipid-laden macrophages, (iv) Culture model #4 (representing pathological intimal thickening), which incorporated multiple layers of human coronary artery smooth muscle cells within the matrix. Coupling this model with different shear stress conditions revealed the effect of low shear stress on the oxidative modification of LDL and the upregulation of pro-inflammatory molecules and matrix-degrading enzymes. Using electron microscopy, immunofluorescence confocal microscopy, protein and mRNA quantification assays, we showed that the behaviors exhibited by the endothelial cells, macrophages and vascular smooth muscle cells in these models were very similar to those exhibited by these

**Funding:** Supported in part by the National Institute of Health (R01 HL144690), Dr. Vincent Miscia Cardiovascular Research Fund, University of Nebraska Collaboration Initiative Seed Grant.

**Competing interests:** "Yiannis S. Chatzizisis: Speaker honoraria, advisory board fees and research grant from Boston Scientific Inc., advisory board fees and research grant from Medtronic Inc., Co-founder of ComKardia Inc. All other authors have no relevant conflict of interests to disclose. This does not alter our adherence to PLOS ONE policies on sharing data and materials."

cell types in nascent and intermediate atherosclerotic plaques in humans. The preparation time of the cultures was 24 hours.

## Conclusion

We present three-cell co-culture models of human atherosclerosis. These models have the potential to allow cost- and time-effective investigations of the mechanobiology of atherosclerosis and new anti-atherosclerotic drug therapies.

## Introduction

Atherosclerosis is a complex molecular and cellular process triggered by low shear stress and evolving from endothelial cells (ECs) dysfunction to subendothelial accumulation of oxidized low-density lipoprotein (LDL), inflammation, and vascular smooth muscle cells (VSMCs) dedifferentiation, transmigration, and proliferation [1–4]. Atherosclerosis develops primarily in arterial curvatures and bifurcations, characterized by low or oscillatory flow, resulting in low time-averaged wall-shear stress ($<10$ dynes/cm$^2$), in the presence of systemic risk factors, including smoking, hypertension, diabetes, hyperlipidemia, and genetic predisposition [4].

Pre-clinical models are of paramount importance for the study of the pathobiology of atherosclerosis and the development of anti-atherosclerotic therapies. Experiments with animal models of atherosclerosis provide important mechanistic insights, however, they cannot reproduce the complex vascular biology and natural history of atherosclerosis that is observed in humans.

Furthermore, experiments on animal models are expensive and time-consuming. *In-vitro* cell culture models of atherosclerosis are more affordable and time-efficient but lack the ability to reproduce the complex molecular pathways and cellular interactions that occur *in-vivo* [5]. The existing cell culture methods in vascular biology fall into three categories: **(i)** Single-cell cultures of ECs, monocytes, or VSMCs on tissue culture plates. These are the most popular, simple, affordable and fast cultures, however, they don't allow the investigation of complex interactions among various cell types and extracellular matrix, [5] **(ii)** Two-cell cultures, which involve two cell types, either co-cultured on top or next to each other (direct co-cultures), or separated by a porous membrane (indirect co-cultures) [5]. These cultures are more complex and cumbersome to build. Even though they are more advanced cultures compared to single-cell cultures, they still lack the ability to reveal the complex multicellular interactions that occur in humans, [3] and **(iii)** Three-cell co-cultures, which are more representative of human vascular biology and allow complex crosstalk among various cell types (direct co-cultures) [6–9]. However, they lack reproducibility, homogeneity, and usually, they require more time, equipment, and cost to develop [5]. Further development of *in-vitro* co-culture models that incorporates human-like vascular wall can enable time- and cost-effective studies of atherosclerosis and facilitate more targeted and meaningful animal and human studies. A cost-effective and highly reproducible three-cell co-culture model of atherosclerosis allowing direct interaction among the major vascular cell types could potentially serve as an attractive tool for pathophysiology and drug studies in atherosclerosis.

The aim of this work was to develop a fast, affordable, and reproducible three-cell co-culture model of the atherosclerotic artery wall that could replicate the different cellular mechanisms of atherosclerosis, extending from atherogenesis to pathological intimal thickening. To date, no mechanistic studies of three-cell co-culture models with direct cell contact have

accounted for the effect of shear stress on the pathophysiology of atherosclerosis. As further validation of our co-culture models, we exposed the co-cultures to varying shear stress conditions encountered in human atherosclerosis and demonstrated that low shear stress significantly augmented the expression of inflammatory mediators associated with atherosclerosis in the presence of LDL, compared to high shear stress and static controls. These findings are in alignment with *in-vivo* studies in patients and animal culture models that showed a strong association between low shear stress and atherosclerosis.

## Materials and methods

### Design of co-culture models

We designed four culture models and the design of this study was illustrated graphically in **Fig 1** (**Graphical abstract**): **(i)** Culture model #1 representing normal arterial intima: [3] In this model, human coronary artery endothelial cells (HCAECs) were cultured on top of collagen

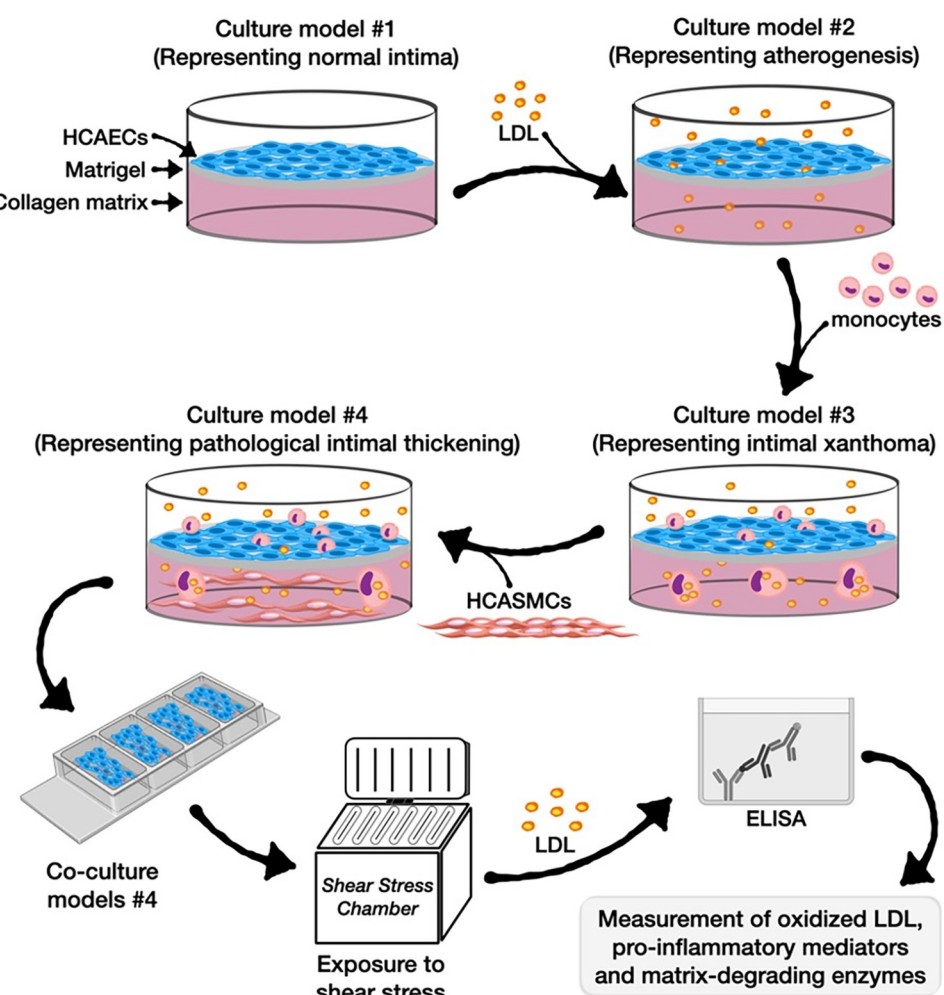

**Fig 1. Graphical illustration of the co-culture models and study design.** HCAECs: human coronary artery endothelial cells; HCASMCs: human coronary artery smooth muscle cells; LDL: low-density lipoproteins.

type I matrix that was pre-coated with solubilized basement membrane matrix (Matrigel; Corning Life Sciences, Durham, NC, USA; **Fig 2A**), **(ii)** Culture model #2, representing atherogenesis: [3] This model was based on the culture model #1 with the addition of native LDL particles labelled with 3,3'-dioctadecylindocarbocyanine (Dil-LDL, 5 μg/mL, Thermo Fischer Scientific, Waltham, MA, USA) in the culture medium (**Fig 3A**), **(iii)** Culture model #3 representing intimal xanthomas or "fatty streaks": [3] This model was based on the culture model #2 with the addition of human THP- 1 monocytes on top of the HCAECs monolayer (**Fig 4A**), **(iv)** Culture model #4 representing pathological intimal thickening: [3] This model was based on the culture model #3 with the addition of multiple layers human coronary artery smooth muscle cells (HCASMCs) inside the collagen matrix (**Fig 5A**).

## Cell preparation

Two batches of healthy HCAECs and HCASMCs were purchased and cultured following the protocols of the cell vendor (Cell Applications, San Diego, CA, USA). Each batch of confluent cells was routinely passaged every 7–8 days by trypsinizing and plating into the tissue culture plate at a density of 0.5–1.0 x $10^6$ cells per 100-mm dish. Cells of passages number #3 to #9 were used for all the co-cultures. Each co-culture model (#1, #2, #3 and #4) was prepared more than 5 times. THP-1 monocytes were cultured in RPMI-1640 medium supplemented with 10% fetal calf serum (10% FCS-RPMI) and passaged every 5–7 days following the protocols of the cell vendor (American Type Culture Collection, Manassas, VA, USA). Fresh human monocytes were purchased from a commercial source (Zei Bio Inc, Durham, NC, USA).

## Collagen type I preparation

Native collagen type I was extracted and purified from rat tail tendon [10, 11]. The tendons were pulled out from rat tails and cut into 1 cm-sized pieces, which were washed three times with tris (hydroxymethyl) aminomethane (Tris)-buffered saline (0.9% NaCl and 10 mM Tris, pH 7.5). Following the wash, the rat tail tendons were dehydrated and sterilized serially with 50%, 75%, 95%, and 100% ethanol. The rat tail tendon collagen was extracted in 6mM HCl at 4°C overnight and purified by centrifuging at 4,000 rpm for 2 hours. The supernatant (pure type I collagen solution in 6mM HCl) was collected and stored at 4°C. To determine the adequate concentration of collagen, a lyophilized aliquot (dehydrated collagen) from each batch of collagen solution was weighed. We ensured the purity of the samples using sodium dodecyl sulfate-polyacrylamide gel electrophoresis, which detected collagen type I and no other detectable proteins. Each batch of rat tail tendon collagen (approximately 600 ml from 4 rat tails) could adequately provide collagen type I for about 120 co-cultures (2–5 ml of rat tail tendon collagen per co-culture).

Live animals were not used in this study. Rat tails were cut from the sacrificed adult rats that were disposed after completing experiments in the Department of Pharmacology and Experimental Neuroscience, University of Nebraska Medical Center. Ethical approval for the rat tail used in this study was waivered by the Institutional Animal Care and Use Committee (IACUC), University of Nebraska Medical Center.

## Collagen matrix preparation

The collagen type I solution was prepared by mixing the rat tail tendon collagen, distilled water and 4 times concentrated DMEM (Dulbecco's Modified Eagle Medium, pH 8.5) on ice. The collagen solution was then polymerized in an incubator at 37°C for 30 minutes to prepare the collagen matrix. The collagen matrix had the following characteristics: (i) pH of 7.40, (ii) Physiological ionic strength, and (iii) Collagen concentration of 0.75 mg/ml [12].

**Fig 2. Culture model #1 representing normal arterial intima. A**: Schematic illustration: Normal HCAECs cultured on top of Matrigel-coated collagen matrix, **B** and **C**: Scanning electron microscopic images showing the ECs monolayer with gap junctions and the underlying fibers of collagen matrix, **D** and **E**: Transmission electron microscopic images showing vertical section of the cultures **F** and **G**: 3D image of the culture by immunofluorescence

confocal microscopy showing the endothelial cell monolayer (blue) connected by VE-cadherin (red) at the gap junctions (ECs: endothelial cells; gj: gap junctions).

## Processing steps of co-culture models

The surface of the collagen type I matrix was coated with Matrigel in an incubator at 37˚C for 1 hour. The rationale for using Matrigel (collagen type IV and laminin) was to represent the basal membrane of ECs. HCAECs (density of $5 \times 10^5$ cells/cm$^2$) were plated on top of the Matrigel-coated collagen matrix and allowed to attach for 2 hours. Additionally, for culture model #3, THP-1 cells (density of $5 \times 10^5$ cells/ cm$^2$) were placed on top of the HCAECs. In culture model #4, HCASMCs (density $1 \times 10^6$ cells/ml) were mixed with the neutralized collagen solution and polymerized for 30 minutes in an incubator. THP-1 cells in culture models #3 and #4 and the human monocytes in culture #4 were allowed to migrate into the collagen matrix for 24 hours.

## Electron microscopy

Detailed structural assessment of the HCAECs, monocytes/macrophages, and HCASMCs was performed with electron microscopy. The collagen solution (rat tail tendon collagen, distilled water and DMEM) with or without HCASMCs was plated in a 48-well plate (250 μL/well). The culture models were fixed with 2% glutaraldehyde and 2% paraformaldehyde in 0.1 M phosphate-buffered saline. For scanning electron microscopy (SEM), the culture models were dehydrated in graded ethanol and dried with the ECs facing up on aluminum stubs in hexamethyldisilazane (Polysciences, Inc. Warrington, PA, USA), sputter-coated with a layer of gold and examined with microscope at 15 kV. For transmission electron microscopy (TEM), the culture models were fixed as described above, dehydrated with ethanol, passed through propylene oxide and embedded in epoxy resin. Ultra-thin sections were cut with an ultramicrotome, counterstained with uranyl acetate and lead citrate, and examined with an electron microscope.

## Confocal microscopy

The visualization of cells (HCAECs, monocytes/macrophages, HCASMCs), and assessment of LDL uptake was carried out using non-immunofluorescence and immunofluorescence confocal microscopy. For the non-immunofluorescence microscopy, cells were pre-stained with calcein AM (2 μM) for 10 minutes. The culture models were then plated in 35-mm glass-bottom microwell dishes (Cat#: P35G-1.5-14-C, MatTek, Ashland, MA; 100μL/well). For the immunofluorescence microscopy, the plated co-culture models were cultured for 24 hours, fixed with cold methanol (-20˚C for 20 min), washed with PBS (phosphate buffered saline), and blocked with 5% donkey serum dissolved in PBS. Subsequently, the cultures were incubated at 4˚C overnight with three primary antibodies: **(i)** Goat anti-VE-cadherin (R&D Systems, Cat#: AF938, 1:500) for ECs, **(ii)** Rabbit anti-CD68 (Abcam, Cat#: ab125047, 1:500) for macrophages and **(iii)** Mouse anti-α-smooth muscle actin (R&D Systems, Cat#: MAB1420,1:500). After 24 hours, dye-conjugated secondary antibodies (Donkey anti-mouse/rabbit/goat-Alexa 488- or 594, Invitrogen, Cat#: A32766, A32814, A32790, A32744, or A32754, 1:2000) were added for 1 hour and nuclei were stained with 4′,6 diamidino-2-phenylindole (DAPI) for 10 minutes.

## Isolation of the HCAECS and HCASMCs in the co-cultures, and immunoblotting of the cell biomarkers

To assess the expression of cell-specific proteins in the co-cultures, we applied immunoblotting. The culture models were plated in a 6-well plate (2 ml/well). Collagen matrix was digested

## Co-Culture Model #2 (Representing Atherogenesis)

**Fig 3. Co-culture model #2 representing atherogenesis. A**: Schematic illustration: Addition LDL particles on the monolayer of the HCAECs cultured on top of Matrigel-coated collagen matrix, **B**: Scanning electron microscopic images, **C** and **D**: Transmission electron microscopic images. Note the presence of lipid droplets within the ECs cytoplasm, **E, F** and **G**: 3D image of the co-culture by non-immunofluorescence confocal microscopy, showing the transportation of the LDL particles (red) from the medium to the subendothelial space through the pre-stained HCAECs monolayer (green and blue), **H**: Presence of oxidized LDL in the medium in a native LDL concentration-dependent manner (EC: endothelial cells; gj: gap junctions; Mit: mitochondria; L: low-density lipoproteins; LS: lysosome). $^*p < 0.05$, $^{**}p < 0.01$.

## Co-culture Model #3 (Representing Intimal Xanthomas)

**Fig 4. Co-culture model #3 representing intimal xanthomas. A**: Schematic illustration: Addition of native LDL and monocytes to the monolayer of HCAECs cultured on top of Matrigel-coated collagen matrix, **B-D**: Scanning electron microscopic images showing the characteristic membrane changes of monocytes adhering on the ECs (**B, C**), and transmigrating through the gap junctions into the subendothelial space (**D**), **E** and **F**: Magnified transmission electron microscopic images showing the characteristic appearance of subendothelial macrophage with eccentric nucleus, villous plasma projections (white asterisks) and lipid droplets, **G-I**: Non-immunofluorescence confocal microscopy with pre-stained monocytes (green). Note the presence of LDL-laden macrophages in the matrix (**I**), **J**: Presence of oxidized LDL in the medium in a native LDL concentration-dependent manner (EC: endothelial cells; gj: gap junctions; L: low-density lipoproteins; Mφ: macrophages; N: nucleus). **$p < 0.01$.

with collagenase I (Sigma cat#: C-0773, 300U/mL serum-free DMEM, 2 collagen gels in 1.0 mL of the collagenase, incubated for 1 hour). Cell pellets of HCAECS and HCASMCs were suspended with 0.5 ml trypsin-EDTA and incubated for 5 minutes followed by inhibiting the trypsin activity with 0.5 ml 10% FCS-DMEM. Cells were then pelleted and re-suspended with 80 μL serum-free-DMEM (SF-DMEM) followed by adding a 30 μL blocking reagent and anti-CD31 microbeads to isolate the HCASMCs from HCAECs. HCASMCs were isolated by

**Fig 5. Co-culture model #4 representing pathological intimal thickening. A**: Schematic illustration: Addition of LDL and monocytes on top of HCAECs cultured on Matrigel-coated matrix containing layers of HCASMCs, **B** and **C**: Scanning electron microscopic images showing the transportation stages of monocytes: Monocytes adhering on the ECs (**B**), transmigrating monocytes through gap junctions (**C**), and transmigrant monocytes (white asterisks in **B**), **D-F**: Transmission electron microscopic images showing monocytes adhering on the ECs (**D**), and layers of spindle-shaped VSMCs in the collagen matrix (**E**) with characteristic surface connections with neighboring collagen fibers (black arrows in **F**), **G-I**: 3D images of the co-culture by non- immunofluorescence confocal microscopy showing the

monolayer of ECs on top of the matrix, LDL (red) and pre-stained monocytes (green) on top of the ECs, LDL-laden macrophages in the matrix (merged green and red), and pre-stained spindle-shaped smooth muscle cells (green), **J**: High magnification of immunofluorescence confocal microscopy showing α-smooth muscle actin stained VSMCs (green), **K**: Immunoblotting of cell-specific markers in HCAECs and HCASMCs, isolated from the co-cultures by magnetic beads. Note the expression of CD31 predominantly by the HCAECs and α-smooth muscle actin predominantly by HCASMCs, **L**: Presence of oxidized LDL in the medium and matrix in a native LDL concentration-dependent manner (EC: endothelial cells; N: nucleus; L: low-density lipoprotein; gj: gap junctions; Mφ: macrophages; VSMC: vascular smooth muscle cells; SMA: α-smooth muscle actin). $^{***}p < 0.001$.

running through the LS column, which was designed for positive selection of the endothelial cells expressing a specific surface biomarker, CD-31 (MACS Milteny Biotec, Auburn, CA, USA). Total proteins of the isolated HCASMCs and HCAECs were subjected to electrophoresis. Proteins were transferred to the PVDF membrane and incubated with primary anti-human CD31 antibody (Cell Application, Cat # CB13678, 1:1000) or anti-α-smooth muscle actin (R&D Systems, Cat#: MAB1420, 1:1000) at four˚C overnight. After washing with buffer the next day, secondary Horseradish Peroxidase-conjugated anti-rabbit (Rockland Immunochemicals, Cat#: 611–1302, 1:2000) or anti-mouse (Rockland Immunochemicals, Cat#: 610–1202, 1:2000) antibodies were applied at room temperature for 1 hour. After washing and developing with an enhanced chemiluminescent substrate (SuperSignal™ West Atto Ultimate Sensitivity Substrate), images of the immunoblotting band were obtained (Kindle Biosciences, LLC, USA).

## Exposure of co-culture model #4 to shear stress

To assess the ability of our cultures to mechanotransduce the shear stress, we pre-exposed the co-culture model #4 to laminar flow of varying shear stress magnitude (no flow, low shear stress: $5 \pm 3$ dynes/cm$^2$, and high shear stress: $30 \pm 3$ dynes/cm$^2$) using a commercially available device (Streamer®; FlexCell International, Burlington, NC, USA) for 1 hour [13]. The cultures pre-exposed to varying shear stress conditions were then transferred to 24-well plates and treated with varying concentrations of native LDL (i.e., 0, 5, 25, and 100 μg/mL; Lee Biosolutions Inc., Maryland Heights, MO) for additional 6 hours.

## Measurement of oxidized LDL

In culture models #2, #3 and #4, 24 hours after the addition of native LDL, the culture medium was collected. The remaining collagen matrix was digested with collagenase and the supernatant was harvested. Oxidized LDL was quantified in the medium and solubilized collagen matrix using a commercially available enzyme-linked immunoassay kit (ELISA; Mercodia AB, Sweden).

## Measurement of chemokine, cytokines, cathepsins, and matrix metalloproteinases (MMPs)

In the medium of culture model #4, we measured the released chemokines [monocyte chemoattractant protein (MCP)-1], cytokines [interleukin (IL)-1ß, IL-6, IL-8,], MMP-1 and -9, and cathepsins L and S using the DuoSet™ ELISA Development Systems kits (Bio-techne, Minneapolis, MN, USA).

## Statistical analyses

Statistical analyses were performed using PRISM 8 (GraphPad, San Diego, CA, USA). Each ELISA, immunoblotting, and real-time RT-PCR experiment was performed at least three

times and, in each time, the ELISA and real-time RT-PCR samples were quantified and averaged across three wells. Values were expressed as means ± standard error of mean. Comparison among groups was performed using a two-way ANOVA, followed by Tukey's test to adjust for multiple comparisons between groups. *P* values of less than 0.05 were considered to be significant.

# Results

## Morphology and function of culture model #1 (representing normal arterial intima)

SEM and TEM showed a single cell layer of HCAECs attached to the underlying Matrigel-coated collagen matrix (**Fig 2B–2E**). Note the gap junctions between adjacent ECs under higher magnification (**Fig 2D** and **2E**). Confocal immunostaining showed the presence of HCAECs on the surface of culture model #1 (**Fig 2F**), as well as the ability of the ECs to express VE-cadherin at the gap junctions (**Fig 2G**). Immunoblotting of the isolated HCAECs from the underlying collagen matrix demonstrated the expression of CD31 by the ECs (**Fig 5K**).

## Morphology and function of culture model #2 (Representing atherogenesis)

Similar to model #1, SEM showed a single layer of HCAECs on the surface of the co-culture (**Fig 3B**). TEM showed the presence of LDL particles in the cytoplasm of the ECs (**Fig 3C** and **3D**). Confocal microscopy of culture model #2 showed dye-labeled LDL (Dil-LDL, red) in the cytoplasm of the ECs (**Fig 3E** and **3F**), as well as in the collagen matrix (**Fig 3G**), suggesting the ability of our model to replicate the LDL uptake by ECs and sub-endothelial accumulation of LDL.

## Morphology and function of culture model #3 (representing intimal xanthomas)

In culture model #3, SEM showed THP-1 monocytes exhibiting a variety of plasma membrane configurations and adhering to the HCAECs monolayer (**Fig 4B–4D**). Also, SEM showed monocytes transmigrating through the EC gap junctions (**Fig 4D**). Monocyte transmigration into the subendothelial matrix was further confirmed by TEM (**Fig 4E**). TEM also showed the transformation of monocytes to macrophages with elongated nuclei, microvillous projections of the cell membrane, and LDL droplets, suggestive of the ability of macrophages to phagocytize LDL and transform to foam cells (**Fig 4E** and **4F**). Non-immunofluorescence confocal microscopy demonstrated the co-localization of dye-labeled LDL (red) with HCAECs (blue nuclei) and pre-stained THP-1 monocytes (green) on the surface of the culture (**Fig 4G** and **4H**), as well as co-localization of LDL with macrophages (orange) in the matrix layer (**Fig 4I**). After 24 h culture, immunofluorescence confocal microscopy was performed using anti-CD68 antibodies targeting macrophages (**Fig 6**). Note, the abundance of CD68-positive macrophages (green) in the matrix (**Fig 6A**), as well as the co-localization of CD68-positive macrophages (green) with dye-labeled LDL (red; **Fig 6B**), suggesting the ability of our culture to promote differentiation of monocytes to macrophages inside the collagen matrix, and subsequent LDL phagocytosis by the macrophages. To visualize the ox-LDL internalization by monocytes/macrophages, we added exogenous Dil-labeled ox-LDL into the medium of model #4. As shown in **Fig 6C**, there was a co-localization of oxidized LDL by monocytes/macrophages and SMCs.

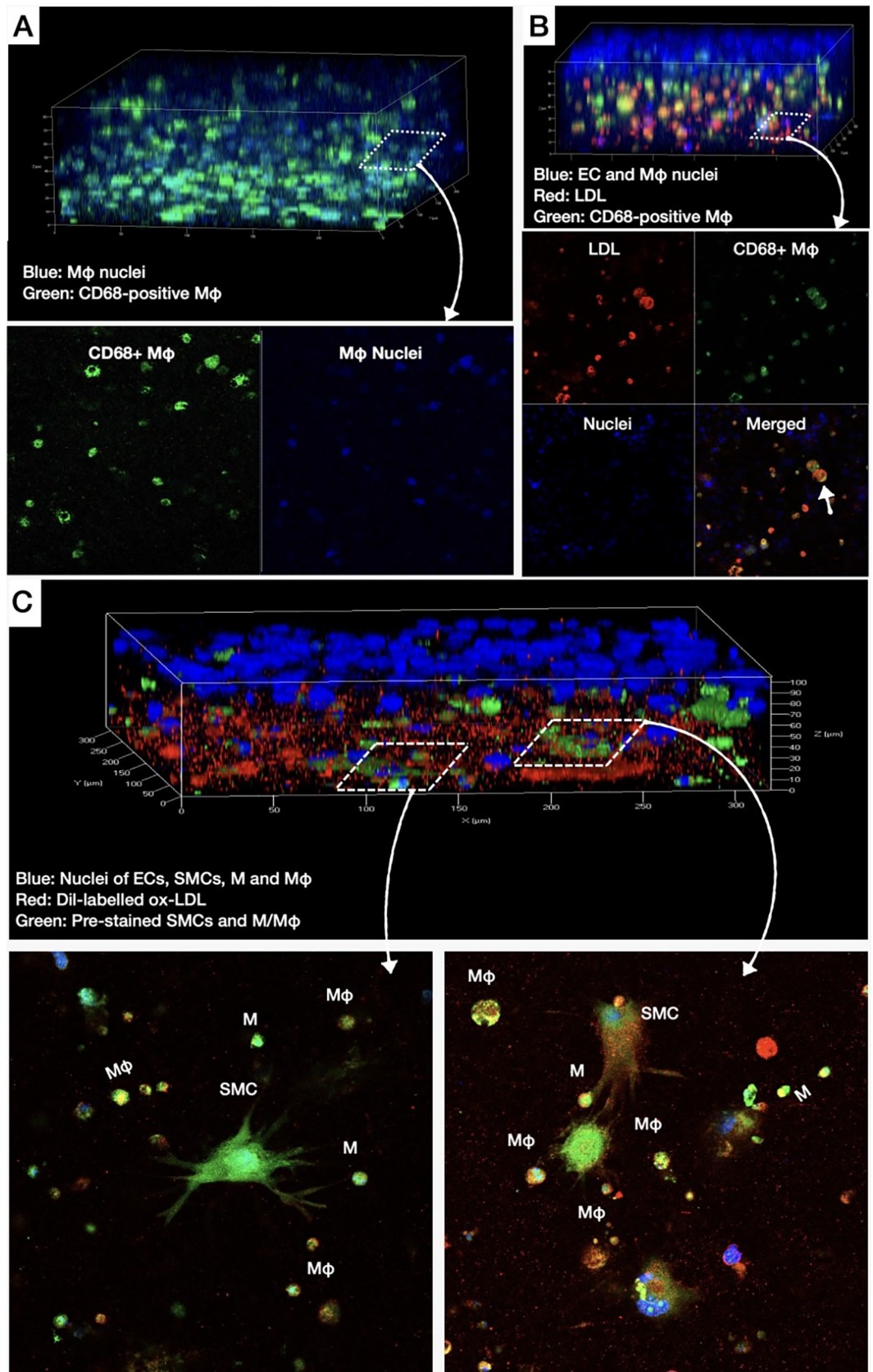

**Fig 6. Differentiation of monocytes into lipid-laden macrophage cells in the collagen matrix. A**: 3D confocal immunofluorescence image of monocytes mixed within collagen matrix and cultured for 24 h, showing their transformation into CD68-positive macrophages (green), **B**: 3D confocal immunofluorescence image of monocytes cultured within EC-plated collagen matrix followed by the addition of Dil-labelled LDL (red). Note the transformation of monocytes into LDL-laden, CD68-positive macrophages (white arrow; merged green and red; Mφ: macrophages; LDL: low-density lipoproteins), **C:** 3D confocal non-immunofluorescence (monocytes and SMCs were pre-stained with calcein AM) image of monocytes/macrophages and SMCs cultured within EC-plated collagen matrix followed by the addition of Dil-labelled ox-LDL (red). Note the co-localization of ox-LDL with SMCs and monocytes/macrophages (M: monocytes; Mφ: macrophages; SMC: smooth muscle cells.

## Morphology and function of culture model #4 (representing pathological intimal thickening)

Similar to culture model #3, SEM and TEM of culture model #4 showed monocyte adhesion and migration (**Fig 5B–5D**). Furthermore, TEM showed layers of spindled-shaped HCASMCs in the collagen matrix (**Fig 5E**) with their characteristic connections between their surface and the neighboring collagen matrix (black arrows, **Fig 5F**). Confocal *microscopy* demonstrated the migration of pre-stained human THP-1 monocytes (green) into the matrix, and transformation into macrophages up taking red-labeled LDL (**Fig 5G–5I**). Similar patterns of transmigration and transformation into macrophages were observed with *human* monocytes (**S1 Fig**). Also, confocal microscopy demonstrated α-smooth muscle actin-stained VSMCs (green) in the collagen matrix (**Fig 5J**). Immunoblotting of the cells using magnetic beads to separate HCAECs from HCASMCs showed the ability of ECs to express CD31, and VSMCs to express α- smooth muscle actin (**Fig 5K**).

The total setup time for co-culture models was about 24 hours, including polymerization of collagen matrix and preparation of the co-culture models by coating Matrigel as well as adding cells and LDL. **Table 1** summarized the setup time of the three-cell co-cultures of the existing three-cell co-culture methodologies, clearly highlighting the time efficiency of our methodology.

## Oxidative modification of LDL

ELISA analysis showed the presence of oxidized LDL in the culture medium of co-culture models #2 and #3, which was directly proportional to the dose of the added native (non-oxidized) LDL, suggesting the ability of the ECs and monocytes to oxidize the LDL (**Figs 3H and 4J**). Furthermore, oxidized LDL was present in both the collagen matrix and medium of co-culture model #4 (**Fig 5L**). Taken together, these results show the ability of our co-cultures to promote the oxidative modification of LDL, which is a key element in the pathophysiology of atherosclerosis.

**Table 1. Comparison of preparation time of three-cell co-culture models of atherosclerosis.**

| Three-cell culture models | Preparation times | | | |
|---|---|---|---|---|
| | *Matrix* | *Smooth muscle cells* | *ECs* | *THP-1 cells/monocytes* |
| Current work | *30 minutes* | *30 minutes* | *2 hours* | *1 day* |
| Dorweiler B, et al. | *2 weeks* | *2 weeks* | *Maximum duration of 4–6 weeks* | *3 days* |
| Navab M, et al. | *2.5 hours* | *2 days* | *3 hours* | *Unknown* |
| Takaku M, et al. | *4 hours* | *7 days* | *7 days* | *5 minutes* |
| Gu X, et al. | *Unknown* | *1 day* | *1 day* | *2 days* |
| Noonan J, et al. | *Unknown* | *2–3 days* | *1 day* | *2 hours* |
| Li Y, et al. | *Unknown* | *3–6 days* | *3–6 days* | *20 hours* |
| Tull SP, et al. | *Unknown* | *1 day* | *1 day* | *Unknown* |

## Effect of shear stress on oxidized LDL, pro-inflammatory cytokines, and matrix-degrading enzymes

Model #4 cultures pre-exposed to low shear stress exhibited a significantly higher concentration of oxidized LDL in the medium compared to the cultures pre-exposed to no flow or high shear stress (**Fig 7A**). Similarly, the pre-exposure to low shear stress significantly increased the release of chemokines (MCP-1), cytokines (IL-1ß, IL-6), cathepsins L, and MMP-1, in the medium, and this effect was augmented by adding LDL in the culture medium (**Fig 7B–7F**). Notably, the low shear stress effect on the concentration of oxidized LDL in both medium and matrix, as well as the expression of chemokines, cytokines, MMPs and cathepsins was augmented with increased doses of native LDL (**Fig 8A–8H**), suggesting the synergistic effect

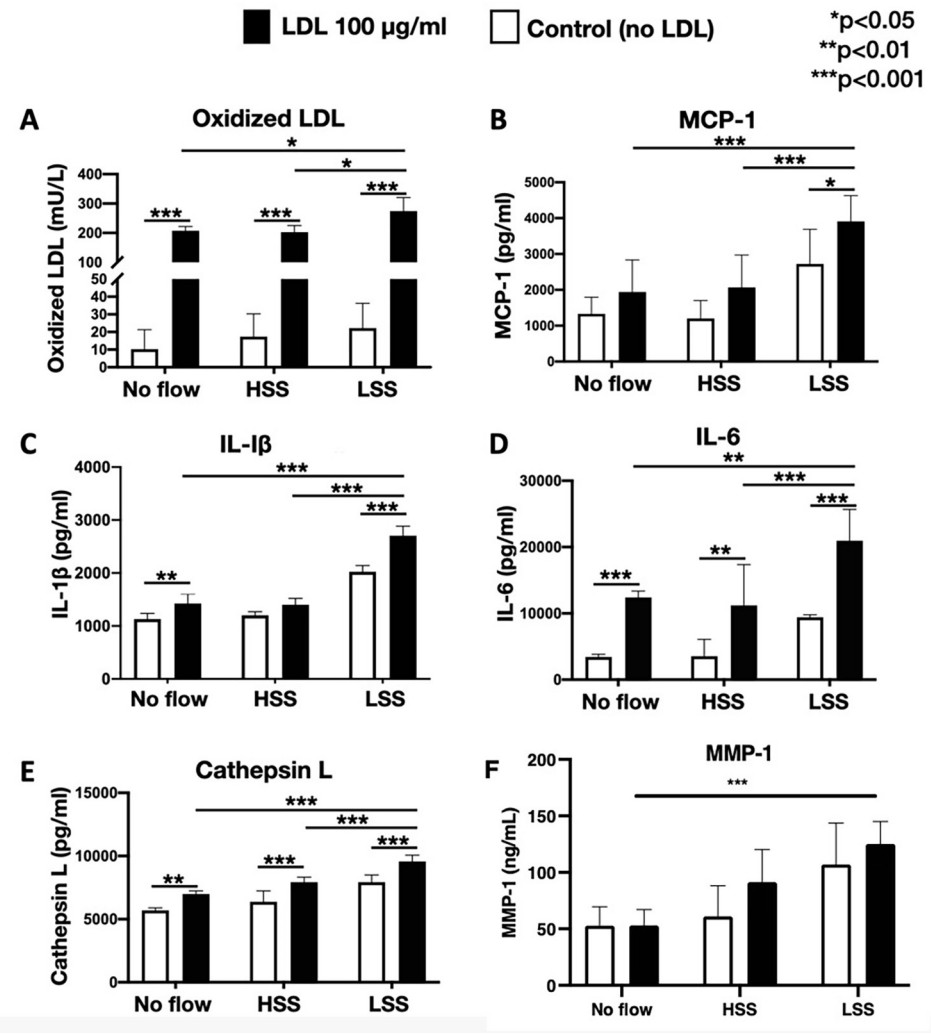

**Fig 7. Effects of shear stress and native LDL on the release of oxidized LDL, expression of pro-inflammatory molecules, and matrix-degrading enzymes. A**: Low shear stress significantly increased the expression of oxidized LDL in the co-culture medium of model #4, compared to high shear stress or static controls, **B-F:** Similarly, expression of MCP-1, IL-1ß, IL-6, cathepsin L, and MMP-1 was increased in cultures exposed to low shear stress, compared to high shear stress or static controls (white bars). Notably, addition of LDL in the medium significantly augmented the expression of oxidized LDL, MCP-1, IL-1ß, IL-6, cathepsin L, and MMP-1 (black bars; LDL: low-density lipoproteins; MCP-1: monocyte chemoattractant protein 1; IL: interleukin; MMP: matrix metalloproteinases) $^*p < 0.05$; $^{**}p < 0.01$; $^{***}p < 0.001$.

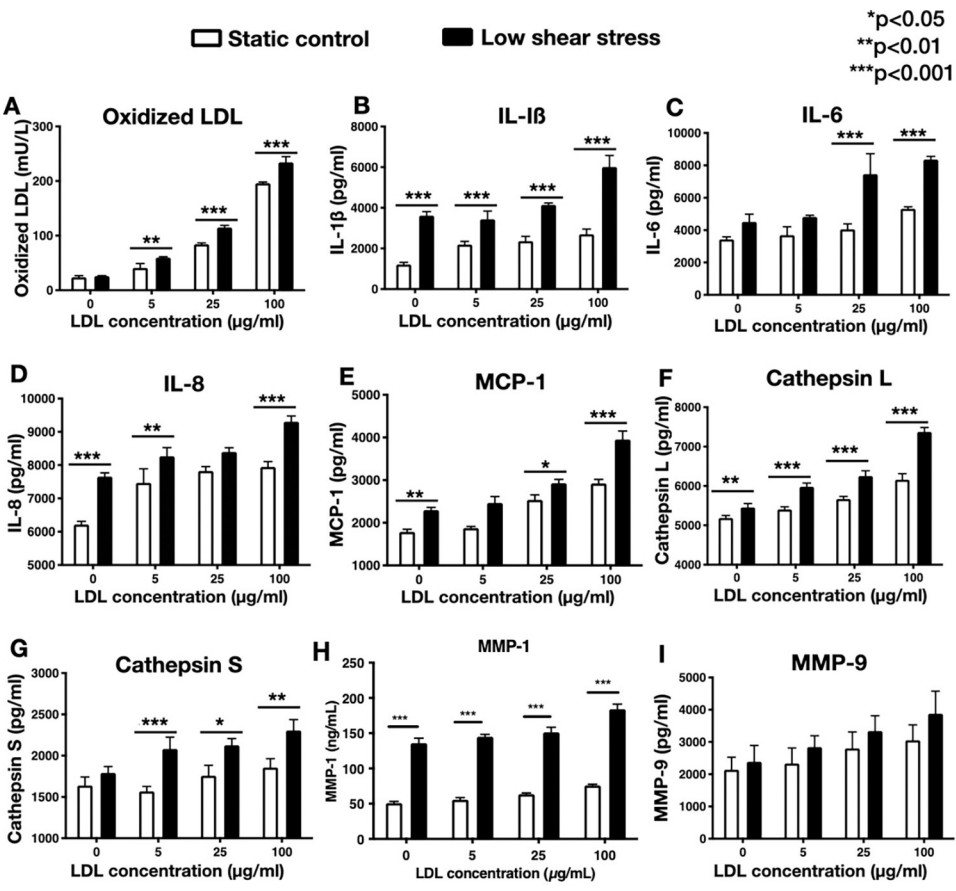

**Fig 8. Effects of low shear stress and native LDL on oxidized-LDL and expression of pro-inflammatory molecules and matrix-degrading enzymes.** Dose-response effect of native (non-oxidized) LDL on the quantities of oxidized LDL (A), MCP-1, IL-1ß, IL-6, IL-8, MMP-1 and -9, and cathepsins L and S (B-I), released into the medium of co-culture model #4 (white bars). Note that the pro-atherosclerotic effect of native LDL is significantly augmented by exposing the co-cultures to low shear stress (5±3 dynes/cm$^2$) for 1 hour, prior to the addition of native LDL (black bars; ox-LDL: oxidized low-density lipoproteins, MCP: monocyte chemoattractant protein; IL: interleukin; MMP: matrix metalloproteinases) * $p<0.05$; ** $p < 0.01$; *** $p<0.001$.

between the pro-atherosclerotic stimulus of low shear stress and LDL. The release of MMP-9 under low shear stress was numerically higher than the no flow across all LDL concentrations, however, the difference was not statistically significant (**Fig 8I**).

## Discussion

In this study, we presented the methodology to construct four different co-culture models of ECs, monocytes/macrophages, and VSMCs, capable of reproducing the different cellular mechanisms of atherosclerosis, [3] extending from normal intima (culture model #1) to atherogenesis (co-culture model #2), intimal xanthomas (co-culture model #3) and pathological intimal thickening (co-culture model #4). In our cultures we incorporated a vascular wall-mimicking extracellular matrix substrate, which consisted of a thin layer of collagen type IV and laminin (representing basal membrane), overlaying a thick matrix of collagen type I (representing subendothelial space). Furthermore, our co-cultures showed a unique ability to mechano-sense the shear stress and promote subendothelial accumulation and oxidative modification of native LDL. In contrast to prior work in the field, these co-culture models had

## Innovative aspects of the co-cultures

In our culture, we used THP-1 cells, which represent highly proliferative monocytes. Even though THP-1 cells are not typical blood cell monocytes, they exhibited similar patterns of endothelial attachment and subendothelial accumulation with blood monocytes (**S1 Fig**). Although several studies have reported the two-cell co-cultures (usually a combination of ECs and VSMCs) as an *in-vitro* model of the arterial wall, only a few of them have investigated aspects of atherosclerosis (**Table 2**). Most of these studies either seeded cells on top or next to each other on tissue culture plates allowing direct intercellular contact, [5, 14] or layered two cell types on the opposite sides of a porous membrane (e.g. Transwell insert) [15]. More co-culture models were also developed by plating ECs (rabbit, human aortic, or human umbilical vein) on top of three-dimensional scaffolds, pre-mixed with multilayered VSMCs, and adding human monocytes that transmigrated in the subendothelial space, particularly in the presence of chemoattractant molecules, [6, 7] oxidized LDL, [7, 8] native LDL [9] or pro-inflammatory mediators [14] (**Table 2**). Once in the subendothelial space, monocytes were shown to differentiate into macrophages, and in the presence of oxidized LDL, formed foam cells [7, 8]. These co-culture models also demonstrated the potential role of neutrophils in atherosclerosis by showing increased transmigration into the subendothelial space with increasing concentrations of LDL [9].

Our study extends the state-of-the-art in three important ways, summarized in **Table 2**. First, we significantly reduced the time needed to create the co-culture models, including collagen matrix preparation and plating of the co-culture models with Matrigel, THP-1 cells, and addition of LDL, from 2–6 weeks reported in previous studies to 24 hours. [6, 9, 16]. **Table 1** summarizes the setup time of the three-cell co-cultures of the existing three-cell co-culture methodologies, clearly highlighting the time efficiency of our methodology. The faster processing of our co-cultures was mostly attributed to the fast preparation of a extracellular matrix substrate, which represented the vascular wall basal membrane (collagen type IV and laminin) and subendothelial space (rat tail tendon collagen type I) [10, 11].

Second, our co-cultures resembled *in-vitro* the early-to-intermediate stages of the natural history of atherosclerosis, as previously described by Virmani R, et al.: [3] **(i)** Normal intima-mimicking cultures, where HCAECs formed a continuous monolayer with gap junctions on top of a Matrigel-coated collagen matrix. The ECs exhibited a human-like phenotype and expressed EC- specific markers (VE-cadherin and CD31), **(ii)** Atherogenesis-mimicking cultures, where native (non-oxidized) LDL particles added in the culture medium were accumulated in the subendothelial space, and underwent oxidative modification (**Fig 3H**). Even though the transportation of native LDL through the EC barrier was not clearly delineated in our cultures, it is likely that LDL passed through the intercellular junctions, as well as through the ECs (transcytosis), given the presence of LDL droplets in the ECs cytoplasm (**Fig 3D**), [17] **(iii)** Intimal xanthoma-mimicking co-cultures, where monocytes added on top of the ECs monolayer underwent plasma membrane changes, attached to the ECs, transmigrated through the intercellular junctions into the subendothelial space, and underwent a transformation to human-like macrophages with characteristic morphological changes (elongated nuclei, cytoplasmic membrane irregularities with microvillous projections [7]) and expression of specific markers (CD68; **Fig 6**). Notably, the macrophages demonstrated the ability to phagocytize LDL and transform it into foam cells (**Fig 4F**). Even though we could not directly visualize the presence of oxidized LDL in the macrophages and that a similar amount of oxidized LDL was

**Table 2. Qualitative comparison of two- and three-cell co-culture models of atherosclerosis.**

| Cell types | Species | Type of added LDL | Ability to oxidize LDL | Exposure to LSS | Transformation of monocytes to macrophages | Time to build culture | Aim |
|---|---|---|---|---|---|---|---|
| **Three-cell co-cultures** | | | | | | | |
| EC-SMC- Mo [6] | Human aortic cells | Native | No | No | Yes | 1 week | To study monocyte and LDL kinetics |
| EC-SMC- Mo [7] | Rabbit cells | Oxidized | No | No | Yes | 3 weeks | Quantitative analysis of monocyte/macrophage transmigration and foam cell formation |
| EC-SMC- Mo [9] | Human umbilical cells | Native | No | No | Yes | 2-4weeks | To show lipid accumulation, monocyte migration, and foam cell formation |
| EC-SMC- Mo [19] | Human aortic | Native and oxidized | Yes | No | Yes | Unknown | To investigate the mechanism of foam cell formation and inhibition by atorvastatin |
| EC-SMC- Mo [20] | Human coronary artery cells | Unknown | No | No | No | 3–6 days | To demonstrate immune-vascular interplay in atherosclerosis |
| EC-SMC- Mo [16] | Human umbilical cells | Oxidized | No | No | No | 1 week | To investigate the inflammatory markers in atherosclerosis |
| EC-SMC-Platelets [21] | Human umbilical cells | Unknown | No | No | No | 3–5 weeks | To demonstrate platelet adhesion to ECs in activated by SMCs |
| EC-SMC- Mo **(Current work, direct contact)** | Human coronary artery cells | Native | Yes | Yes | Yes | 24 hours | Co-culture models mimicking early to intermediate stages of atherosclerosis and coupled with shear stress, to study the mechanobiology of atherosclerosis |
| **Two-cell co-cultures** | | | | | | | |
| EC-SMC (Indirect contact) [15] | Bovine cells | Native | No | Yes | No | 48 hours | SMCs transmigration and drug testing |
| EC-SMC (Indirect contact) [22] | Canine | Unknown | No | No | No | 2 days | To study fibroblast growth factor-2-toxin induced cytotoxicity |
| EC-SMC (Direct contact) [23] | Calf ECs and rat SMCs | Unknown | No | No | No | 48 hours | To study ECs migration and transformation from SMCs-derived angiogenic factors |
| EC-SMC (Indirect contact) [24] | Human umbilical cells | Unknown | No | Yes | No | 3–4 days | To study the phenotypic changes in ECs and SMCs on exposure to LSS |
| EC-SMC (Indirect/ direct contact) [25] | Human umbilical cells | Native | No | No | No | 10 days | To study the lipid infiltration/accumulations |
| EC-SMC (Indirect contact) [26] | Rat cells | Unknown | No | Yes | No | Unknown | To study the effect of LSS on cell migration and proliferation |
| EC-SMC (Indirect contact) [27] | Human umbilical cells | Unknown | Unknown | Yes | No | 4 weeks | To elucidate the role of miR-126 secreted by endothelial cells (ECs) in regulating SMC turnover |
| EC-SMC (Direct contact) [14] | Human umbilical cells | Unknown | No | No | No | 3 days | To determine the anti-coagulation, anti-hyperplasia and inflammatory markers |

ECs: endothelial cell; SMCs: smooth muscle cells; Mo: monocytes; LDL: low density lipoproteins; ox-LDL: oxidized LDL; LSS: low shear stress

found in the absence (**Fig 3H**, model #2) or presence (**Fig 4J**, model #3) of monocytes, the fact that the macrophages contained LDL droplets, and oxidized LDL was identified in the matrix of the cultures, suggested that the macrophages in our co-cultures were capable of up-taking oxidized LDL. **(iv)** Pathological intimal thickening-mimicking cultures, where multiple layers

of spindle-shaped VSMCs were mixed within the collagen matrix followed by plating ECs on top of the matrix. Monocytes were then plated on top of the EC monolayer and allowed to migrate into the subendothelial collagen matrix. Using this model, we examined the effect of LDL and low shear stress on the production of pro-atherogenic molecules and found that the LDL was oxidized in an LDL concentration-dependent manner (12.5–50 μg/mL native LDL). Notably, oxidation of LDL was dramatically augmented in model #4 (ECs, SMCs and monocytes exposed to low shear stress; **Fig 5L**) compared to model #2 (ECs only; **Fig 3H**) or model #3 (ECs and monocytes; **Fig 4J**), suggesting the multiplying effect of the low shear stress on the LDL oxidation and that low shear stress further augmented LDL oxidation. Also, the oxidation of LDL was increased in model #4 (ECs, SMCs, and monocytes, **Fig 5L**) compared to model #3 (ECs and monocytes, **Fig 4J**), even under no flow (control) conditions, suggesting that the presence of SMCs plays a role in the augmented LDL oxidization. In addition, LDL significantly promoted the release of pro-inflammatory chemokines and cytokines (MCP-1, IL-1ß, IL-6, and IL-8) and matrix-degrading enzymes (cathepsin L, S, MMP-1, -9) which are thought to play an important role in foam cell formation and apoptosis—key steps in the development of a necrotic core and plaque progression to advanced phenotypes [2]. In addition, low shear stress further enhanced the effect of LDL stimulation on the release of the aforementioned pro-inflammatory molecules and enzymes. Overall, these findings revealed the unique ability of our co-cultures to replicate key pathobiological processes of human atherosclerosis.

Third, our model was coupled with the pro-atherogenic stimulus of low shear stress (similar to the shear stress encountered in human atherosclerosis), which, especially in the presence of increasing doses of native (non-oxidized) LDL, promoted the oxidative modification of LDL, and augmented the expression of pro-inflammatory cytokines and matrix-degrading enzymes. Of note, the source of LDL oxidization appeared to be the ECs given that the amount of oxidized LDL between model #2 (ECs only) and #3 (ECs and monocytes) were comparable. This finding is in line with the literature [4].

Finally, our model incorporated human coronary artery ECs, VSMCs, and human monocytes, in contrast to the previous studies, which used ECs and VSMCs taken from animals, human aortic or human umbilical veins, [6, 9] therefore, our co-culture models appear to mimic the pathophysiology of human coronary artery atherosclerosis.

## Future perspectives

*In-vitro* representation of normal vascular wall and atherosclerotic plaque by co-culturing ECs, monocytes, and VSMCs is a promising approach to investigating the pathogenesis and treating various cardiovascular diseases, including atherosclerosis. In the current study, we developed human-like three-cell co-culture models coupled with shear stress. This is the first time that a co-culture model with direct three-cell contact has the combination of the following important elements of atherosclerosis: (i) Human coronary artery cells, (ii) Exposure to varying shear stress conditions varying flow conditions, (iii) Native LDL, and (iv) Ability to promote LDL oxidation. These novel features along with the fast processing time make our cultures a very compelling system for time- and cost-effective *in-vitro* investigation of new hypotheses in vascular biology of atherosclerosis, [4, 18] in-depth molecular signaling analyses, and new anti-atherosclerotic drugs. Overall, our proposed co-culture models have the potential to guide more focused animal and clinical studies.

## Limitations

There were several limitations within the current study. First, the collagen matrix used was extracted from rat tail tendons. Even though the matrix in our co-culture models contained

collagen type I and IV, further modification of the matrix by adding fibronectin and proteogly-cans would potentially make it more representative of the human vascular wall. The addition of the ECs on top of collagen matrix appeared to mimic structurally and functionally the normal intima; however, we acknowledge that the normal intima is more complex. Second, the number of HCASMCs ($10^6$ cells/mL) may not be representative of the population of VSMCs in the vascular walls under normal or pathologic conditions. Third, we did not directly visualize the presence of oxidized LDL in the macrophages, but, as discussed in the results, we believe that the abundant lipid droplets found in the cytoplasm of macrophages were most likely representing oxidized LDL. Fourth, the physiological concentration of LDL (1 mg/ml) showed massive stimulation of chemokines, cytokines, and matrix-degrading enzymes (**S2 Fig**). However, since smaller concentrations of LDL (0.05–0.1 mg/ml) showed comparable results, we elected to use smaller doses of LDL in our experiments. Fifth, in the current work, we primarily assessed the expression of pro-inflammatory cytokines, chemokines, and matrix-degrading enzymes and did not measure the protease activity. Sixth, our cultures developed macrophages in the stage of transformation to foam cells, but not typical foam cells. We believe that by prolonging the culture time by a few more days, the macrophages would potentially evolve into typical foam cells or even apoptotic foam cells. Seventh, in our co-culture models, the quantification of the number of monocytes that transmigrated across the EC layer and infiltrated into the matrix was technically challenging. Our approximate estimation is 5–6 monocytes per 1,000 monocytes attached on the ECs.

## Conclusions

In the current study, we presented a fast, cost-effective, and reproducible co-culture model that can replicate the different cellular mechanisms of early and intermediate stages of atherosclerosis. Most importantly, our models were coupled with shear stress allowing the study of the mechanobiology of atherosclerosis. The behaviors exhibited by the ECs, macrophages, and VSMCs in these models were very similar to those exhibited by these cell types in nascent and intermediate atherosclerotic plaques *in-vivo*. Future work will be focused on further evolving the co-culture models to mimic fibroatheromas with well-developed lipid pools and necrotic cores, rupture- or erosion-prone phenotypes, and calcifications.

## Supporting information

**S1 Fig. Confocal microscope image of co-culture model #4 with monocytes isolated from human blood. A. Three-dimensional image of the co-culture.** Confocal microscope observation on the co-culture of HCASMC, HCAEC, and monocytes from human peripheral blood. **B. Images of sub-endothelial layer.** Note the Dil-LDL (red), monocytes (green, pre-stained with calcein AM), and endothelial cells (nuclei in blue) at the top of the 3D images. **C. Images of smooth muscle cell (SMC) layer.** Note the spindle-shaped HCASMC (green, pre-stained with calcein AM) as well as the nuclei (blue) of the cells in the bottom of the 3D co-culture. (TIF)

**S2 Fig. Effect of higher concentration of LDL (1.0 mg/mL) and shear stress on release of oxidized-LDL, pro-inflammatory cytokines, and enzymes. A.** Native-LDL concentration-dependent release of oxidized LDL. **B-F:** Release of MCP-1, IL-1ß, IL-6, cathepsin L, and MMP-1 was increased in cultures exposed to low shear stress, and moreover, in the presence of higher concentrations of LDL, release of pro-inflammatory cytokines and enzymes was further augmented (closed bar). LDL: low-density lipoprotein; MCP-1: monocyte chemoattractant protein- 1; IL: interleukin; MMP: matrix metalloproteinase. $^*p<0.05$; $^{**}p<0.01$;

$^{***}p<0.001$.
(TIF)

**S3 Fig. Original images of immunoblotting of the cells using magnetic beads to separate HCAECs from HCASMCs showed the ability of ECs to express CD31, and VSMCs to express F061—smooth muscle actin.** HCAEC: human coronary artery endothelial cell; HCASMC: human coronary artery smooth muscle cell.
(TIF)

## Author Contributions

**Conceptualization:** Yiannis S. Chatzizisis.

**Data curation:** Martin Liu, Saurabhi Samant, Charu Hasini Vasa, Usama M. Oguz, Daniel R. Anderson.

**Formal analysis:** Martin Liu, Sangjin Ryu, Timothy Wei.

**Funding acquisition:** Yiannis S. Chatzizisis.

**Investigation:** Saurabhi Samant, Ryan M. Pedrigi, Devendra K. Agrawal, Yiannis S. Chatzizisis.

**Methodology:** Martin Liu, Saurabhi Samant, Ryan M. Pedrigi, Devendra K. Agrawal, Yiannis S. Chatzizisis.

**Project administration:** Ryan M. Pedrigi, Yiannis S. Chatzizisis.

**Resources:** Martin Liu, Daniel R. Anderson, Yiannis S. Chatzizisis.

**Supervision:** Martin Liu, Yiannis S. Chatzizisis.

**Validation:** Martin Liu, Saurabhi Samant, Ryan M. Pedrigi, Sangjin Ryu, Timothy Wei, Yiannis S. Chatzizisis.

**Visualization:** Martin Liu, Yiannis S. Chatzizisis.

**Writing – original draft:** Yiannis S. Chatzizisis.

**Writing – review & editing:** Charu Hasini Vasa, Usama M. Oguz, Yiannis S. Chatzizisis.

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
