## [Decision Letter · Decision Letter 0]

7 Oct 2022

PONE-D-22-25235Co-culture models of endothelial cells, macrophages, and vascular smooth muscle cells for the study if the natural history of atherosclerosisPLOS ONE

Dear Dr. Chatzizisis,

Thank you for submitting your manuscript to PLOS ONE. After careful consideration, we feel that it has merit but does not fully meet PLOS ONE’s publication criteria as it currently stands. Therefore, we invite you to submit a revised version of the manuscript that addresses the points raised during the review process.

The proposed article describing cellular models of the different stages of atherosclerosis is of interest to the journal. However, the relevance of these models to describe the stages of atherosclerosis needs to be further developed and better argued. In addition, the reviewers have highlighted many points that need to be clarified and completed. Thank you for answering point by point to their questions and remarks.

We look forward to receiving your revised manuscript.

Kind regards,

Alain-Pierre Gadeau, Ph.D

Academic Editor

PLOS ONE

“Yiannis S. Chatzizisis: Speaker honoraria, advisory board fees and research grant from Boston Scientific Inc., advisory board fees and research grant from Medtronic Inc., Co-founder of ComKardia Inc. All other authors have no relevant conflict of interests to disclose.”

Reviewers' comments:

Reviewer's Responses to Questions

**Comments to the Author**

1. Is the manuscript technically sound, and do the data support the conclusions?

Reviewer #1: Partly

Reviewer #2: Partly

2. Has the statistical analysis been performed appropriately and rigorously? 

Reviewer #1: Yes

Reviewer #2: N/A

3. Have the authors made all data underlying the findings in their manuscript fully available?

Reviewer #1: Yes

Reviewer #2: Yes

4. Is the manuscript presented in an intelligible fashion and written in standard English?

Reviewer #1: Yes

Reviewer #2: Yes

5. Review Comments to the Author

Reviewer #1: In this manuscript by Liu et al, the authors have developed four different co-culture models in order to in vitro replicate different stages of early atherosclerosis: model 1 for normal arterial intima, model 2 for atherogenesis, model 3 for intima xanthomas (mimicking "fatty streaks"), and model 4 for pathological intima thickening. Each model was characterized by electron (SEM and TEM) and confocal microscopies together with measurements of oxidized LDL and expression of chemokines, cytokines and matrix proteinases. Model 4 was also subjected to pre-exposition to low shear stress in order to mimic what is encountered in human atherosclerosis.

Even if the proposed co-culture models suffer from several limitations that are clearly mentioned by the authors at the end of the manuscript (pages 26-27), these models show improvements compared to those already published in the literature and are expected to help researchers working in the field. I recommend acceptation of this work after addressing the following comments.

1) Figure numbering must be corrected. Figure 1 should contain the graphical illustration of the co-culture models and study design but is missing. The current Figure 1 corresponds to the culture model 1 (indicated as Figure 2 in the manuscript). As a result, the numberings of the figures are shifted.

2) Figure relative to model 2: higher magnification/better resolution is recommended for confocal imaging to better show colocalization of LDL particles and endothelial cells.

3) Figure relative to model 2: what was the concentration of native LDL used? 0.1 and 1.0 mg/dL or 0.1 and 1.0 mg/mL (like for model 3)? This should be confirmed.

4) It would have been interesting to evaluate the permeability of the EC monolayer before and after adding LDL or monocytes?

5) If comparable concentrations of native LDL were added in the medium for model 2 and model 3 (see previous comment), does it means that adding monocytes in the model has no influence on the oxidization of native LDL? This should be discussed by the authors.

6) What is the percentage of monocytes that transmigrate through the EC monolayer for models 3 and 4? Is it comparable?

7) Figure relative to model 4: for the western blot profile (panel K), HCASMCs exhibit similar amount of SMA compared to HCASMCs in monoculture. Authors should check other marker proteins in order to better characterize the phenotype of the SMC (contractile versus synthetic). Is there production of ECM by SMC in this model?

8) Figure relative to model 4: to be homogeneous with figures relative to models 2 and 3, "LDL (µg/mL)" in the legend of panel L should be replaced by "native LDL added in the medium (µg/mL)".

9) Why authors did not used comparable concentrations of native LDL in the medium for model 4 (as for model 2 and 3)? This should be explained by the authors.

10) M&M section:

- page 9: authors indicated that HCAECs and HCASMCs were used between passages 3 and 9. Have the authors checked the phenotype of the cells between passage 3 and 9? Unchanged?

- page 9: preparation of type I collagen should be better explained. After extraction of collagen, any precipitation and/or dialysis of collagen? What about the presence of HCl?

- page 12: authors should explain what does the LS column means?

11) page 19: Writing "the LSS effect on the concentration of oxidized LDL in both medium and matrix, as well as the expression of chemokines, cytokines, MMPs and cathepsins was directly proportional to the dose of added native LDL (Fig. 8)" is not correct. For instance, no variation was observed between 5 and 25 µg/mL LDL and between static and flow conditions for Il-1beta, Il-8, CathS, L, MMP1 and MMP9. The authors should modify the text accordingly.

12) Authors should comment on the differences observed for the concentrations measured in the medium+matrix (Fig. 6/7?) versus medium alone (Fig. 7/8?) for the different cytokines, chemokines, MMPs, … at 100µg/mL LDL and under low shear stress. For instance, 150-200 pg/mL MMP-1 in the medium (fig. 7/8?) versus 100.000-150.000 pg/mL in medium+matrix ; around 6.000 pg/mL IL-1b in the medium around 3.000 pg/mL for medium+ matrix.

Reviewer #2: The aim of this work is to set up a cell co-culture system to represent the different stages of atherosclerosis. Although the idea of establishing in vitro cell culture system to mimic atherosclerosis is of interest, there are several limitations in this study.

-in my opinion presenting the different models as representative stages of atherosclerosis development is irrelevant. For example, mixing endothelial cells with collagen is not representative of healthy vessel. They should focus only on their model 4 as in vitro co-culture system to “mimic atherosclerosis”.

-the authors provided images but they should quantify them (e.g. monocyte transmigration, foam cells…) to have an idea of the frequency of the processes

-quantifying oxLDL is useless, they should instead visualize oxLDL internalization into the macrophages

-page 22, fig 5 results are missing

-fig 3j , the quantification of oxldl is the same as in fig 2h. In this regard, why adding ldl at 0.1 mg/ml did not give the same result as in fig 2? Anyway, as said before this quantification is useless.

6. PLOS authors have the option to publish the peer review history of their article (what does this mean?). If published, this will include your full peer review and any attached files.

Reviewer #1: No

Reviewer #2: No

---

## [Author Response · Author response to Decision Letter 0]

25 Nov 2022

Reviewer 1: We would like to thank the Reviewer for the critique. We provided detailed clarifications and revisions based on the Reviewer suggestions and comments.

Reviewer 2: We would like to thank the Reviewer for the critique. We provided detailed clarifications and revisions based on the Reviewer suggestions and comments.

---

## [Decision Letter · Decision Letter 1]

19 Dec 2022

PONE-D-22-25235R1Co-culture models of endothelial cells, macrophages, and vascular smooth muscle cells for the study if the natural history of atherosclerosisPLOS ONE

Dear Dr. Chatzizisis,

Thank you for submitting your manuscript to PLOS ONE. After careful consideration, we feel that it has merit but does not fully meet PLOS ONE’s publication criteria as it currently stands. Therefore, we invite you to submit a revised version of the manuscript that addresses the points raised during the review process.

The revised document is improved. However, there are still several points that need to be strengthened, in particular some that reviewer 2 had raised:

1) It is stated that the models represent “morphologically and functionally the different histopathologic stages of early atherosclerosis”. It would be more accurate to say that they represent “different cellular mechanisms of early atherosclerosis” as they do not take into account many other cell types and cellular and matrix organisation involved in vessel wall during the atherosclerosis process.

2) I understand the challenge of doing an absolute count of macrophages. However, could you provide some relative information on the aproximative frequency of observation of macrophages having crossed the endothelial layer in order to indicate to a future user of the models if this is a rare or easy to observe event? For example giving average macrophage number per confocal field you observed / size of observed confocal field.

3) You show that LDL oxidation is strongly increased in model 4 in the presence of SMC in no flow in comparison to model 3. If so, is there a role for SMC in LDL oxidation.

We look forward to receiving your revised manuscript.

Kind regards,

Alain-Pierre Gadeau, Ph.D

Academic Editor

PLOS ONE

Reviewers' comments:

Reviewer's Responses to Questions

**Comments to the Author**

1. If the authors have adequately addressed your comments raised in a previous round of review and you feel that this manuscript is now acceptable for publication, you may indicate that here to bypass the “Comments to the Author” section, enter your conflict of interest statement in the “Confidential to Editor” section, and submit your "Accept" recommendation.

Reviewer #1: All comments have been addressed

Reviewer #2: (No Response)

2. Is the manuscript technically sound, and do the data support the conclusions?

Reviewer #1: (No Response)

Reviewer #2: Partly

3. Has the statistical analysis been performed appropriately and rigorously? 

Reviewer #1: (No Response)

Reviewer #2: N/A

4. Have the authors made all data underlying the findings in their manuscript fully available?

Reviewer #1: (No Response)

Reviewer #2: (No Response)

5. Is the manuscript presented in an intelligible fashion and written in standard English?

Reviewer #1: (No Response)

Reviewer #2: Yes

6. Review Comments to the Author

Reviewer #1: (No Response)

Reviewer #2: The authors addressed some of my concerns but others remain as following:

-comment 1: even that the authors claim that ECs were mixed with collagen in a “sophisticated way “, this does not mean that it is representative of an atherosclerosis stage. In my opinion, this part should be removed from the manuscript as it does not represent an atherosclerotic stage, neither a “healthy vessel”.

- Comment 2: the authors provided images of monocyte transmigration.I have asked to provide a quantification of this phenomenon, however the authors estimated that this is too challenging. I do not understand the challenging part of this quantification. In case that the infiltration is rare, the authors should acknowledge it.

7. PLOS authors have the option to publish the peer review history of their article (what does this mean?). If published, this will include your full peer review and any attached files.

Reviewer #1: No

Reviewer #2: No

---

## [Author Response · Author response to Decision Letter 1]

23 Dec 2022

Reviewer 1: No response

Reviewer 2: We would like to thank the Reviewer for the comments. We provided detailed clarifications and revisions based on the Reviewer suggestions and comments.

---

## [Editor Report · Decision Letter 2]

28 Dec 2022

Co-culture models of endothelial cells, macrophages, and vascular smooth muscle cells for the study if the natural history of atherosclerosis

PONE-D-22-25235R2

Dear Dr. Chatzizisis,

We’re pleased to inform you that your manuscript has been judged scientifically suitable for publication and will be formally accepted for publication once it meets all outstanding technical requirements.

Kind regards,

Alain-Pierre Gadeau, Ph.D

Academic Editor

PLOS ONE
---

## [Editor Report · Acceptance letter]

5 Jan 2023

PONE-D-22-25235R2 

Co-culture models of endothelial cells, macrophages, and vascular smooth muscle cells for the study of the natural history of atherosclerosis 

Dear Dr. Chatzizisis:

I'm pleased to inform you that your manuscript has been deemed suitable for publication in PLOS ONE. Congratulations! Your manuscript is now with our production department. 

Kind regards, 

on behalf of

Dr Alain-Pierre Gadeau 

Academic Editor

PLOS ONE